# Joint Optimal Power Allocation and Relay Selection Scheme in Energy Harvesting Two-Way Relaying Network

**Xin Song** [1,2], **Siyang Xu** [1,2,*], **Zhigang Xie** [1,2] **and Xiuwei Han** [1,2] 

[1] Engineering Optimization and Smart Antenna Institute, Northeastern University, Qinhuangdao 066004, China; sxin78916@neuq.edu.cn (X.S.); Zhigangxie@foxmail.com (Z.X.); xiuweihan@stumail.neu.edu.cn (X.H.)

[2] School of Computer Science and Engineering, Northeastern University, Shenyang 110819, China

[*] Correspondence: 1670738@stu.neu.edu.cn

**Abstract:** In this paper, we propose a joint power allocation, time switching (TS) factor and relay selection scheme for an energy harvesting two-way relaying communication network (TWRN), where two transceivers exchange information with the help of a wireless-powered relay. By exploiting the TS architecture at the relay node, the relay node needs to use additional time slots for energy transmission, reducing the transmission rate. Thus, we propose a joint resource allocation algorithm to maximize the max-min bidirectional instantaneous information rate. To solve the original non-convex optimization problem, the objective function is decomposed into three sub-problems and solved sequentially. The closed-form solution of the transmit power of two sources and the optimal TS factor can be obtained by the information rate balancing technology and the proposed time allocation scheme, respectively. At last, the optimal relay node can be obtained. Simulation results show that the performance of the proposed algorithm is better than the traditional schemes and power-splitting (PS) scheme.

**Keywords:** energy harvesting; two-way relaying network; max-min bidirectional information rate; power allocation; relay selection

---

## 1. Introduction

Recently, cooperative communication has become an effective method to achieve spatial diversity by establishing a virtual multi-antenna structure. In a cooperative communication system, relay is used to assist the communication from the source node to its far-end destination node. Due to the communication nodes operated at a half-duplex mode, the information exchanging between two nodes in the one-way relaying network requires four time slots. Thanks to the physical network coding technology, two-way relaying network (TWRN) can complete the bidirectional information transmission in two phases, which effectively mitigates the loss of spectral efficiency [1–3].

Although cooperative communication has many advantages, the node power constraint is still a tricky problem. Since relay nodes are always placed in harsh or dangerous places, recharging or replacing batteries of relay nodes may be inconvenient and can incur a high cost [4]. Energy harvesting technology has recently emerged as a promising approach to prolong the lifetime of energy-constrained wireless network from ambient energy, such as wind, solar, etc. However, the received energy at nodes is random and intermittent, which motivates an upsurge of research interest on radio frequency (RF)-based simultaneously wireless information and power transfer [5–7]. For RF energy harvesting relay nodes, there are two major energy harvesting relay protocols, which are named as "time switching

relay (TSR) protocol" and "power splitting relay (PSR) protocol". For the TSR protocol, the receiver switches two statuses between information decoding and energy harvesting. For the PSR protocol, the receiver splits the signal into two streams of a different power for providing the transmission power and decoding information separately to complete the whole transmission. In one study [8], a dynamic PS scheme was proposed to split the received signal with adjustable power levels based on the instantaneous channel condition, which could obtain the optimal PS rule at the receiver to achieve various trade-offs between the maximum ergodic capacity and the maximum average harvested energy. For another performance standard, outage probability was investigated in PSR asymmetric two-way relay system [9]. In addition, the system outage performance was studied when the channel state information was imperfect [10]. Additionally, a multi-relay cooperation system could be used to improve the sum rate with PSR protocol [11]. Thus, some paper propose that PSR theoretically achieves better performance [12]. However, PS requires appropriate power split circuits to decide how much of the signal is harvested as energy and how much of it is used for extracting information. This in turn, increases the complexity/cost of the hardware. On the other hand, hardware non-idealities can significantly reduce the efficiency of PS technology [13]. Therefore, TS can be widely investigated due to its low implementation complexity [14,15]. In another study [16], the throughput of the TSR one-way relay system was investigated. In this paper, we employ the TS protocol in TWRN.

The TSR relay nodes need to use additional time slots for energy transmission, which will reduce the performance of the system. Relay selection and power allocation are two effective ways to improve system performance in the TWRN. Traditional relay selection scheme is based on the channel status information between the relay and the user [17]. In one study [18], Bayes theorem was applied in the relay selection scheme, which improved the throughput significantly. Combining power allocation, the algorithm in another study [19] selected the relay with minimum symbols error probabilities as the cooperative relay and obtained the closed-form expression of the power. In one finding [20], a joint relay selection and power allocation for TWRN was proposed to maximize the smaller of the received SNRs of the two transceivers under the total transmitted power budget. Also, the two novel power allocation schemes have been found to maximize the upper bound of sum rate and could achieve the trade-off of outage probability between two terminals [21]. For maximizing the energy efficiency, the optimal power allocation and relay selection scheme can be obtained [22]. In [23], the power allocation and relay selection algorithm is designed by energy pricing in the decode-and-forward cooperative network. Due to the introduction of energy harvesting technology, the traditional power allocation and relay selection algorithms will no longer be applicable for novel systems. Therefore, this paper proposed a joint power allocation and relay selection algorithm for the TSR relay network.

In this paper, we consider an AF-based two-way relay network with energy harvesting technology, where two sources exchange information by the TSR energy harvesting relay node. We first combine the TSR protocol with the two-way relaying system and propose an optimal power allocation and relay selection scheme, which aims to maximize max-min bidirectional instantaneous information rate. The closed-form expressions of the optimal power with the fixed TS factor and the fixed relay node can be obtained when the bidirectional link information rate is equal. Then, a time allocation scheme is used to obtain the optimal TS factor, which can maximize information rate with the optimal transmit power and the fixed relay node. Lastly, the optimal relay can be obtained with the optimal transmit power and TS factor. Simulation results show that the proposed power allocation and relay selection scheme outperforms the traditional method and the PS scheme.

The rest of paper is organized as follows. Section 2 introduces the system model of the TSR energy harvesting two-way relaying network. Section 3 presents a joint power allocation, relay selection and TS factor optimization scheme. In Section 4, numerical results verify the better performance of the proposed scheme compared with traditional algorithms. Finally, Section 5 concludes this paper.

## 2. System Model

The considered system model and the TS architecture are shown in Figure 1. One source wants to transfer its own information to a far-end source. Due to the long-distance information transmission, two sources cannot communication directly. Thus, two sources use the selected TS relay node to transmit information and the system model can be abstracted as Figure 1. The abstracted system model consists of two source nodes $(S_1, S_2)$ and a set of $N$ relay nodes $(R_i, i = 1, 2 \ldots N)$. The relay nodes are merely powered through wireless energy transfer from two sources and all relay nodes adopt amplify-and-forward mode. There is no direct link between the two source nodes $S_1$ and $S_2$ due to deep fading. Each node is equipped with a single antenna and operated in a half-duplex mode. $h_i$ and $g_i$ are donated as the instantaneous channel gains between $S_1$ to $R_i$ (or $R_i$ to $S_1$) and $S_2$ to $R_i$ (or $R_i$ to $S_2$), respectively. $h_i$ and $g_i$ are assumed to be independent and distributed complex Gaussian random variables (CGRVs) with zero-mean and variances $\sigma_{hi}^2$ and $\sigma_{gi}^2$. According to TSR protocol, the relay node uses a switch-like structure to acquire energy or signals in two time phases. Thus, the whole transmission block $T$ is divided into $\alpha T$, $(1 - \alpha)T/2$ and $(1 - \alpha)T/2$, where $\alpha \in [0, 1]$ is donated as TS factor. $\alpha T$ part is called energy harvesting phase, $(1 - \alpha)T/2$ part is called multiple-access phase, and the remainder $(1 - \alpha)T/2$ part is called broadcast phase, as shown in Figure 2.

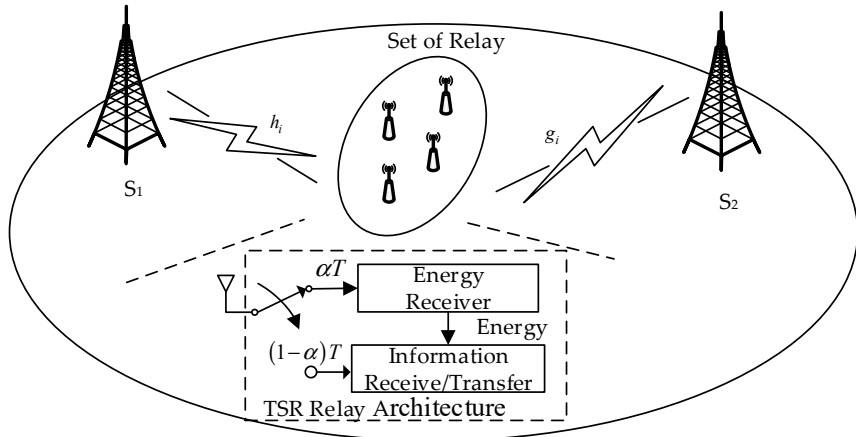

**Figure 1.** Illustration of a two-way AF relay system and the TS relay architecture.

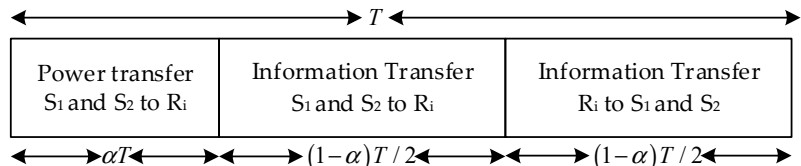

**Figure 2.** TSR protocol at the energy harvesting relay.

In the energy harvesting phase, the relay nodes receive the energy signal from two source nodes $S_1$ and $S_2$ through RF transmission. The signal $y_{R_i,E}$ received by the relay nodes can be written as

$$y_{R_i,E} = \sqrt{P_{S1}}h_i s_1 + \sqrt{P_{S2}}g_i s_2 + n_{R_i} \tag{1}$$

where the parameters $P_{S1}$ and $P_{S2}$ are the transmit power from two sources, and the parameter $n_{R_i} \sim CN(0, \sigma^2)$ is the additive noise. Since the energy of the noise signal $n_{R_i}$ is negligible as compared to that harvested energy from the RF signal [16], the harvested energy $E_{R_i}$ at relay node $R_i$ can be written as

$$E_{R_i} = \eta \alpha T \left( P_{S1}|h_i|^2 + P_{S2}|g_i|^2 \right) \tag{2}$$

where the parameter $\eta(0 < \eta < 1)$ is the conversion efficiency of the energy receiver and $\alpha$ is the TS factor. Then, the received energy is used for the broadcast phase, and we can describe the relay transmit power $P_{R_i}$ as

$$P_{R_i} = \frac{E_{R_i}}{(1-\alpha)/2} = \frac{2\eta(P_{S1}|h_i|^2 + P_{S2}|g_i|^2)\alpha}{1-\alpha} \tag{3}$$

In multiple-access phase, $S_1$ and $S_2$ transmit the information signal $s_1$ and $s_2$ simultaneously to the selected $i$th relay ($R_i$), so the received signal at the relay node $R_i$ can be denoted as

$$y_{R_i} = \sqrt{P_{S1}}h_i s_1 + \sqrt{P_{S2}}g_i s_2 + n_{R_i} \tag{4}$$

In the broadcast phase, the $i$th relay node $R_i$ amplifies the received signal then broadcasts it to two source nodes $S_1$ and $S_2$. The received signal at $S_1$ and $S_2$ can be presented as

$$y_{S1} = h_i(\beta y_{R_i}) + n_1 \tag{5}$$

$$y_{S2} = g_i(\beta y_{R_i}) + n_2 \tag{6}$$

where the parameters $n_1$ and $n_2$ follow $CN(0, \sigma_1^2)$ and $CN(0, \sigma_2^2)$, respectively. The parameter $\beta$ is denoted as the amplifying gain at the relay node, which can be considered by normalization of the received signal, i.e.,

$$\beta = \sqrt{\frac{P_{R_i}}{P_{S1}h_i^2 + P_{S2}g_i^2 + \sigma^2}} \approx \sqrt{\frac{2\eta\alpha}{1-\alpha}} \tag{7}$$

Substituting (4) into (5) and (6), the received signal can be rewritten, respectively,

$$y_{S1} = \beta\sqrt{P_{S1}}h_i^2 s_1 + \beta\sqrt{P_{S2}}h_i g_i s_2 + \beta h_i n_{R_i} + n_1 \tag{8}$$

$$y_{S2} = \beta\sqrt{P_{S2}}g_i^2 s_2 + \beta\sqrt{P_{S1}}h_i g_i s_1 + \beta g_i n_{R_i} + n_2 \tag{9}$$

where $\beta\sqrt{P_{S1}}h_i^2 s_1$ and $\beta\sqrt{P_{S2}}g_i^2 s_2$ are considered as the self-interference of the two sources. We assume that the total channel state information (CSI) can be obtained at two source nodes $S_1$ and $S_2$. Thus, the self-interference can be completely eliminated. After self-interference cancellations of the nodes $S_1$ and $S_2$, (8) and (9) can be rewritten as

$$\widetilde{y}_{S1} = \beta\sqrt{P_{S2}}h_i g_i s_2 + \beta h_i n_{R_i} + n_1 \tag{10}$$

$$\widetilde{y}_{S2} = \beta\sqrt{P_{S1}}h_i g_i s_1 + \beta g_i n_{R_i} + n_2 \tag{11}$$

## 3. Joint Optimal Power Allocation and Relay Selection Scheme

In this section, we propose a joint optimal power allocation, TS factor and relay selection scheme for wireless-powered TWRN that maximizes the smaller of the bidirectional instantaneous information rate under some constraint conditions. As mentioned in [20], the performance of TWRN is mainly affected by the worse instantaneous information rate between $S_1 \to R_i \to S_2$ and $S_2 \to R_i \to S_1$. Therefore, the proposed scheme first optimizes the transmit power. Then the optimal TS factor can be obtained with the optimal power allocation. At last, we select the optimal relay with the maximum instantaneous information rate in the set. Applying (10) and (11), the instantaneous signal-to-noise-ratio (SNR) at two source nodes $S_1$ and $S_2$ can be described as

$$\gamma_{S1} = \frac{\beta^2 P_{S2}|h_i|^2|g_i|^2}{\beta^2|h_i|^2\sigma^2 + \sigma_1^2} = \frac{2\eta\alpha P_{S2}|h_i|^2|g_i|^2}{2\eta\alpha|h_i|^2\sigma^2 + (1-\alpha)\sigma_1^2} \tag{12}$$

$$\gamma_{S2} = \frac{\beta^2 P_{S1} |h_i|^2 |g_i|^2}{\beta^2 |g_i|^2 \sigma^2 + \sigma_2^2} = \frac{2\eta\alpha P_{S1} |h_i|^2 |g_i|^2}{2\eta\alpha |g_i|^2 \sigma^2 + (1-\alpha)\sigma_2^2} \tag{13}$$

Since the information is only transmitted within $(1-\alpha)T$ time and all nodes operated in a half-duplex mode, the instantaneous information rate from $S_1$ to $S_2$ and the instantaneous information rate from $S_2$ to $S_1$ are given by

$$R_{S1} = \frac{1-\alpha}{2} \log_2(1 + \frac{2\eta\alpha P_{S2} |h_i|^2 |g_i|^2}{2\eta\alpha |h_i|^2 \sigma^2 + (1-\alpha)\sigma_1^2}) \tag{14}$$

$$R_{S2} = \frac{1-\alpha}{2} \log_2(1 + \frac{2\eta\alpha P_{S1} |h_i|^2 |g_i|^2}{2\eta\alpha |g_i|^2 \sigma^2 + (1-\alpha)\sigma_2^2}) \tag{15}$$

Therefore, the optimization function of the proposed scheme can be expressed as

$$\begin{aligned}
\operatorname*{argmax}_{i} \ \max_{P_{S1}, P_{S2}, \alpha} \ &\min(R_{S1}, R_{S2}) \\
\text{s. t. } &P_T \leq P_T^{\max} \\
&P_{S1} \geq 0, P_{S2} \geq 0 \\
&0 \leq \alpha \leq 1 \\
&i \in (1, \ldots N)
\end{aligned} \tag{16}$$

where $P_T^{\max}$ is the total transmit power and $P_T = P_{S1} + P_{S2}$. In order to remove the minimized symbol, the original optimization problem can be converted as

$$\begin{aligned}
\operatorname*{argmax}_{i} \ \max_{P_{S1}, P_{S2}, \alpha, t} \ & t \\
\text{s. t. } &R_{S1} \geq t \\
&R_{S2} \geq t \\
&P_T \leq P_T^{\max} \\
&P_{S1} \geq 0, P_{S2} \geq 0 \\
&0 \leq \alpha \leq 1 \\
&i \in (1, \ldots N)
\end{aligned} \tag{17}$$

To obtain the optimal solution of (17), we divide the optimization problem (17) into three parts. First, we perform an optimal power allocation in the case of the fixed TS factor and the fixed relay node. Second, we perform an optimal TS factor with the optimal power allocation and fixed relay node. Last, we perform an optimal relay selection with the optimal power allocation and the optimal TS factor.

For the fixed TS factor and the fixed relay node, the optimization problem can be described as

$$\begin{aligned}
\max_{P_{S1}, P_{S2}, t} \ & t \\
\text{s. t. } &R_{S1} \geq t \\
&R_{S2} \geq t \\
&P_T \leq P_T^{\max} \\
&P_{S1} \geq 0, P_{S2} \geq 0
\end{aligned} \tag{18}$$

**Lemma 1.** *The optimization problem (18) is optimal when $R_{S1} = R_{S2}$, i.e., $\gamma_{S1} = \gamma_{S2}$.*

**Proof of Lemma 1.** The total power constraint $P_T = P_{S1} + P_{S2}$ can be rewritten as $P_{S1} = \lambda P_T$ and $P_{S2} = (1-\lambda)P_T$. It is easy to know that $R_{S1}$ and $R_{S2}$ are the monotonically increasing function and the monotonic decreasing function with respect of $\lambda$, respectively. Also, the objection function is to select the smaller instantaneous information rate of two links. Thus, the optimal power allocation can be obtained when $R_{S1} = R_{S2}$. □

According to **Lemma 1**, we obtain that

$$\frac{2\eta\alpha P_{S1}|h_i|^2|g_i|^2}{2\eta\alpha|g_i|^2\sigma^2 + (1-\alpha)\sigma_2^2} = \frac{2\eta\alpha P_{S2}|h_i|^2|g_i|^2}{2\eta\alpha|h_i|^2\sigma^2 + (1-\alpha)\sigma_1^2} \tag{19}$$

For simple calculation but without loss of generality, we assume $\sigma^2 = \sigma_1^2 = \sigma_2^2$. The Equation (19) can be rewritten as

$$P_{S1}\left(2\eta\alpha|h_i|^2 + (1-\alpha)\right) = P_{S2}\left(2\eta\alpha|g_i|^2 + (1-\alpha)\right) \tag{20}$$

Then, applying (20) into (18), the optimization problem (18) can be written as

$$\max_{P_{S1},t} t$$
$$\text{s.t. } t = \frac{1-\alpha}{2}\log_2\left(1 + \frac{2\eta\alpha P_{S1}|h_i|^2|g_i|^2}{2\eta\alpha|g_i|^2\sigma^2 + (1-\alpha)\sigma^2}\right) \tag{21}$$
$$P_{S1} + \frac{P_{S1}\left(2\eta\alpha|h_i|^2 + (1-\alpha)\right)}{2\eta\alpha|g_i|^2 + (1-\alpha)} \le P_T^{\max}$$

For the objective function (21), $t$ is a monotonically increasing function with respect to $P_{S1}$. It is easy to get the optimal power value $P_{S1}^*$ when $P_T = P_T^{\max}$. And combined with Equation (20), the optimal transmit power of two sources are given by

$$P_{S1}^* = P_T^{\max}\frac{2\eta\alpha|g_i|^2 + (1-\alpha)}{2\eta\alpha|g_i|^2 + 2\eta\alpha|h_i|^2 + 2(1-\alpha)} \tag{22}$$

$$P_{S2}^* = P_T^{\max}\frac{2\eta\alpha|h_i|^2 + (1-\alpha)}{2\eta\alpha|g_i|^2 + 2\eta\alpha|h_i|^2 + 2(1-\alpha)} \tag{23}$$

For optimal power allocation and fixed relay node, the optimization problem can be written as

$$\max_{0\le\alpha\le1} R_T(\alpha) = \frac{1-\alpha}{2}\log_2\left(1 + \frac{2\eta\alpha P_{S1}|h_i|^2|g_i|^2}{2\eta\alpha|g_i|^2\sigma^2 + (1-\alpha)\sigma^2}\right)$$
$$\text{s.t. } P_{S1} = P_{S1}^* \tag{24}$$
$$0 \le \alpha \le 1$$

**Lemma 2.** *$R_T(\alpha)$ is a concave function about $\alpha$ with the range of 0 to 1.*

**Proof of Lemma 2.** We first solve the first derivative of the function $R_T(\alpha)$ with respect of the TS factor $\alpha$.

$$\frac{dR_T(\alpha)}{d\alpha} = -\frac{1}{2}\log\left(1 + \frac{\alpha X}{Y\alpha + 1 - \alpha}\right) + \frac{1-\alpha}{2\ln 2}\left(\frac{X}{(\alpha X + \alpha Y + 1 - \alpha)(Y\alpha + 1 - \alpha)}\right)$$

where $X = \eta P_T^{\max}|h_i|^2|g_i|^2$ and $Y = \eta\alpha|g_i|^2 + \eta\alpha|h_i|^2 + (1-\alpha)$. The second order derivative of $R_T(\alpha)$ with respect to $\alpha$ is given by

$$\frac{dR_T(\alpha)^2}{d^2\alpha} = -\frac{X}{\ln 2}\frac{(1-\alpha) + X(1+\alpha)}{(X\alpha + Y\alpha + 1 - \alpha)(Y\alpha + 1 - \alpha)^2}$$
$$-\frac{X}{\ln 2}\frac{(1-\alpha)(X + Y - 1)(Y\alpha + 1 - \alpha)}{((X\alpha + Y\alpha + 1 - \alpha)(Y\alpha + 1 - \alpha))^2}$$

Since $X \gg 1$ and TS factor $\alpha$ exists in $[0,1]$, we can obtain that $dR_T(\alpha)^2/d^2\alpha < 0$. Thus, there is a optimal $\alpha$ value that maximizes $R_T(\alpha)$.

**Lemma 2** is proved. $\square$

According to **Lemma 2**, the original optimization problem with respect to $\alpha$ is a strictly concave function. Note that $dR_T(\alpha)/d\alpha > 0$ when $\alpha = 0$ and $dR_T(\alpha)/d\alpha < 0$ when $\alpha = 1$. Obviously, there exists an optimal TS factor $\alpha^*$ in $[0, 1]$. However, it is difficult to obtain the closed-form expression of optimal $\alpha^*$ since the first-order derivative of $R_T(\alpha)$ is so complicated and there exist a large amount of unknown. Therefore, the following time allocation scheme is proposed to get the optimal value of $\alpha^*$ when $dR_T(\alpha)/d\alpha = 0$.

---

**Time allocation scheme**

---

1: Initialization: setting the value $\alpha_a = 0$, $\alpha_b = 1$, $\delta$ is a positive real number close to 0;
2: **While** $|\alpha_a - \alpha_b| \geq \delta$ do
3: Based on $dR_T(\alpha)/d\alpha$, calculate $\xi = dR_T(\alpha)/d\alpha$, where $\alpha = (\alpha_a + \alpha_b)/2$;
4: **If** $\xi = 0$, set $\alpha_a = (\alpha_a + \alpha_b)/2$ and go to 9;
5: **Else if** $\xi > 0$, set $\alpha_a = (\alpha_a + \alpha_b)/2$;
6: **Else if** $\xi < 0$, set $\alpha_b = (\alpha_a + \alpha_b)/2$;
7: **end if**
8: **End while**
9: $\alpha^* = \alpha_a$ is the optimal solution.

---

For optimal power allocation and optimal TS factor, the original problem in (16) is reduced to the following relay selection problem:

$$i^* = \arg \max_{P_{S1} = P_{S1}^*, \alpha = \alpha^*, i \in [1,...N]} R_T \qquad (25)$$

The optimal relay selection method can be described by selecting the cooperative relay with the maximum instantaneous information rate in $N$ relays.

The original max-min bidirectional instantaneous information rate can be solved by the proposed joint optimal power allocation and relay selection algorithm.

---

**Joint Optimal Power Allocation and Relay Selection**

---

1: Initialization parameters;
2: Obtain $P_{S1}^*$ and $P_{S2}^*$ for fixed $\alpha$ and fixed relay node following the procedure in Equations (18)–(23);
3: Obtain the optimal $\alpha^*$ for optimal $P_{S1}^*$, $P_{S2}^*$ and fixed relay node using the time allocation scheme;
4: Obtain the optimal cooperative relay node using the traversal method for optimal $P_{S1}^*$, $P_{S2}^*$ and optimal TS factor $\alpha^*$ in $N$ relays;
5: end.

---

## 4. Simulation Results and Analysis

In this section, simulation results are performed to present the performance of the proposed power allocation and relay selection scheme. We assume that there are 10 relays in the set and the energy conversion efficiency is $\eta = 0.95$. Moreover, the distance between $S_1$ and $S_2$ is normalized to unit value, and the distance between $S_1$ to $R_i$ is expressed as $d$. According to the reference [20], the information rate of TWRN is largest when the relay node is located on the midpoint between $S_1$ and $S_2$. Therefore, we assume that $d = 0.5$. Thus, the channel strength of the two channels are $h_i = v_1/d^\alpha$ and $g_i = v_2/(1-d)^\alpha$, where $\alpha$ is the path loss exponent. In this paper, $\alpha$ is set to 2.5. In addition, two source nodes and the relay node have the same noise variance $\sigma^2$. The SNR can be expressed as $\text{SNR} = P_T^{\max}/\sigma^2$. To the Monte Carlo experiment, the simulation results are averaged over 1000 independent channel realizations. Simulation parameters are elaborated in Table 1.

**Table 1.** Simulation parameters.

| Parameter | Value |
| --- | --- |
| Path loss exponent | 2.5 |
| The distance between $S_1$ to $R_i$ | 0.5 |
| The distance between $S_2$ to $R_i$ | 0.5 |
| Energy conversion efficiency | 0.95 |
| Number of relay nodes | 10 |
| Monte Carlo experiment | 1000 |

Figure 3 depicts the information rate curves of using different SNRs under various TS factor $\alpha$. It can be observed that these curves have the same tendency under the different total transmit power constraints. From the trend, the three curves are concave, which demonstrate the accuracy of the analysis in Section 3. As can be seen from Figure 3, the system information rate can achieve 2.67 bps/HZ when the total transmitted power is 15 dB. When the total transmitted power is 20 dB, the system information rate can achieve 3.387 bps/HZ. When the total transmitted power is 25 dB, the system information rate can achieve 4.2 bps/HZ. We note that the optimal TS factor can be found in the range (0, 1).

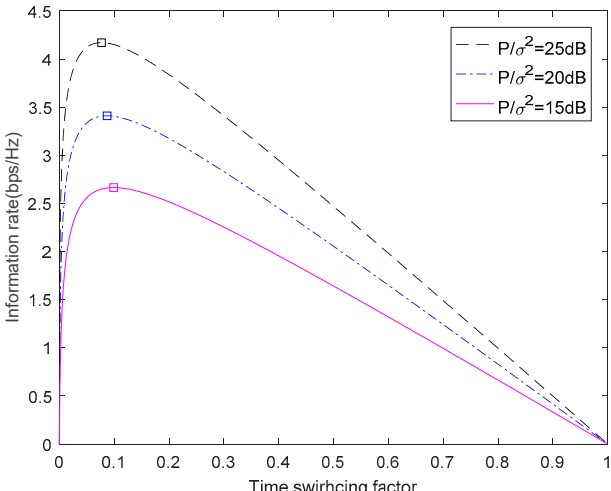

**Figure 3.** The information rate versus TS factor.

The information rate versus SNRin three different schemes is showen in Figure 4. The three schemes are the proposed scheme, the joint resource allocation PS scheme and equal power allocation PS scheme. For the joint resource allocation PS scheme, the power allocation scheme is based on the end-to-end SNRs balancing criterion. The instantaneous SNRs at two source nodes can be formulated as

$$\gamma_{S1} = \frac{\eta\rho(1-\rho)P_{S2}|h_i|^2|g_i|^2}{\left(\eta\rho(1-\rho)|h_i|^2 + \eta\rho|h_i|^2 + (1-\rho)\right)\sigma^2} \tag{26}$$

$$\gamma_{S2} = \frac{\eta\rho(1-\rho)P_{S1}|h_i|^2|g_i|^2}{\left(\eta\rho(1-\rho)|g_i|^2 + \eta\rho|g_i|^2 + (1-\rho)\right)\sigma^2} \tag{27}$$

where $\rho \in [0,1]$ is the PS factor and it is fixed as 0.5. The max-min bidirectional information rate has the maximum when $\gamma_{S1} = \gamma_{S2}$. Therefore, the optimal power allocation scheme under the total power constraint can be expressed as

$$P_{S1} = \frac{\eta\rho(2-\rho)|g_i|^2 + (1-\rho)}{2(1-\rho) + \eta\rho(2-\rho)|g_i|^2 + \eta\rho(2-\rho)|h_i|^2}P_T^{\max} \tag{28}$$

$$P_{S2} = \frac{\eta\rho(2-\rho)|h_i|^2 + (1-\rho)}{2(1-\rho) + \eta\rho(2-\rho)|g_i|^2 + \eta\rho(2-\rho)|h_i|^2} P_T^{\max} \tag{29}$$

Then, we select the cooperative relay node from the relay set, which can maximize the information rate. For the equal power allocation scheme, two source nodes have equal transmission power $P_{S1} = P_{S2} = P_T^{\max}/2$ and the relay selection is random. As shown in Figure 4, the performance of the proposed scheme is better than the other two PS schemes where the SNR ranges from 0 dB to 20 dB. That is because the PS factor is fixed and the cooperative relay is chosen randomly while the proposed scheme has the optimal TS factor, the optimal cooperative relay and the optimal power allocation.

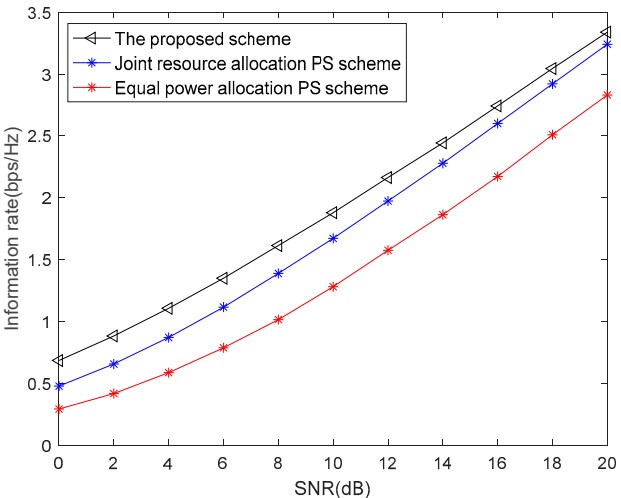

**Figure 4.** The proposed scheme versus the PS scheme.

Figure 5 shows the information rate of our proposed scheme under four different cases. The fix TS factor is set as $\alpha_F = 0.5$. Case 1 is the proposed algorithm that has the optimal relay, the optimal power allocation and the optimal TS factor, which is called O-PA&O-RS&O-TS. The optimal relay, the optimal power allocation and the fixed TS factor is called O-PA&O-RS&F-TS as case 2. Case 3 is the optimal relay, the equal power allocation $P_{S1} = P_{S2} = P_T^{\max}/2$ and the fixed TS factor, which is called E-PA&O-RS&F-TS. Case 4 is the random relay, the equal power allocation and the fixed TS factor, which is called E-PA&R-RS&F-TS. From Figure 5, we note that the optimal relay selection has a great impact on the information rate by comparing case 3 and case 4. However, the information rate of case 2 and case 3 are very similar, for the system only allocates power to two source nodes, and the channels are distributed and independent. As shown in Figure 5, case 4 has the worst performance in four algorithms. The performance of case 2 and case 3 are improved compared with case 4, while the proposed algorithm has the best performance. The information rate of the proposed scheme is about 1.4 bps/HZ higher than case 2 and case 3 when SNR = 20 dB.

Figure 6 displays the information rate versus SNR by comparing the proposed scheme, joint power allocation and relay selection in [20] and the traditional equal power allocation (EPA) scheme. From reference [20], the optimal values of $P_{S1}$, $P_{S2}$ and $P_{R_i}$ can be obtained as

$$P_{S1} = \frac{P_T^{\max}\sqrt{1 + P_T^{\max}|g_i|^2}}{2\sqrt{1 + P_T^{\max}|h_i|^2} + 2\sqrt{1 + P_T^{\max}|g_i|^2}} \tag{30}$$

$$P_{S2} = \frac{P_T^{\max}\sqrt{1 + P_T^{\max}|h_i|^2}}{2\sqrt{1 + P_T^{\max}|h_i|^2} + 2\sqrt{1 + P_T^{\max}|g_i|^2}} \tag{31}$$

$$P_{R_i} = P_T^{\max}/2 \tag{32}$$

With the optimal power allocation solution found, selecting the relay node with maximum SNR as cooperative relay. The traditional EPA scheme is defined as $P_{S1} = P_{S2} = P_{R_i} = P_T^{\max}/3$ and the relay selection is random. From Figure 6, it can be seen that the proposed scheme has better performance than the other two schemes. And compared with the other two schemes, the relay node can harvest energy through wireless–powered, which solve the power constraint of relay nodes.

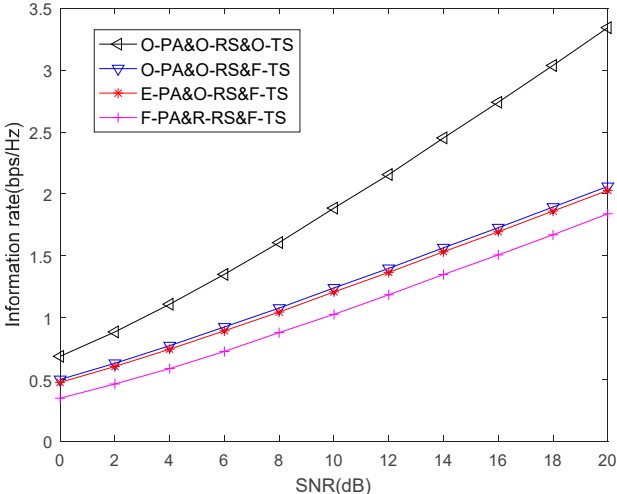

**Figure 5.** Performance of TS scheme in different power allocation and relay selection methods.

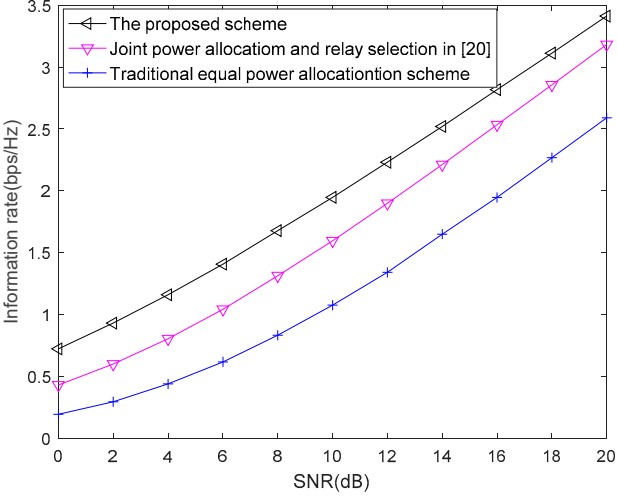

**Figure 6.** Performance comparison between the proposed scheme and the conventional scheme.

As shown in Figure 7, the number of relay nodes in the relay set has an impact on the information rate. The proposed algorithm has the lowest information rate when $R = 10$. As the number of relays increases, the information rate increases. When the number of relays reaches 100, the information rate can reach 3.49 bps/HZ. The improvement of the information rate can reach 0.1 bps/HZ when the relay numbers from 10 to 50. However, the improvement of the information rate is only about 0.02 bps/HZ when the relay numbers from 50 to 100, which indicates that excessive candidate relays do not significantly improve the information rate. In future research, we can expand the optimal number of candidate relays in this study.

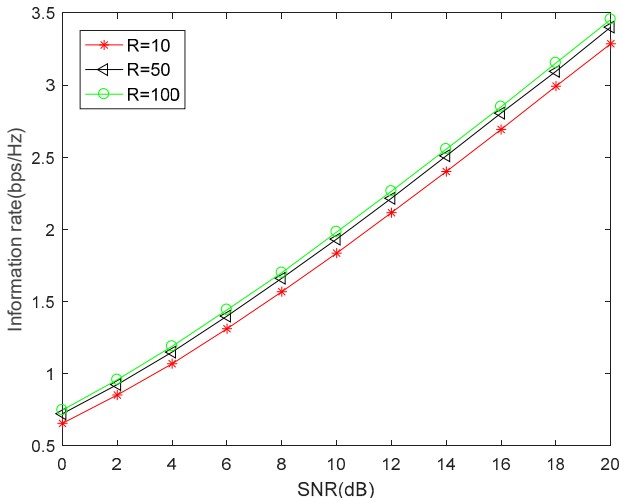

**Figure 7.** Information rate versus different relay number in the relay set.

## 5. Conclusions

In this paper, we propose a joint power allocation and relay selection algorithm based on TS relay architecture, which aims to maximize the max-min bidirectional instantaneous information rate. The system model can be widely applied into the fields of 5G, wireless sensor, internet of things, cognitive radio and wireless body area network. By using the balancing information rate criterion, the closed-form of the optimal power allocation can be obtained. Then, the proposed time allocation scheme can get the optimal TS factor. At last, the relay with the largest instantaneous information rate of the set is selected as the optimal cooperative relay. Energy harvesting technology can solve the energy problem in energy-constrained nodes and reflect the concepts of green environmental protection and sustainable development. The proposed algorithm can make up the loss of the information rate for energy transmission. Simulation results show that the proposed scheme provides effective performance improvement over the traditional schemes. However, energy harvesting technology may cause loss of spectral efficiency. Therefore, combining full-duplex technology will be the direction for future research.

**Author Contributions:** Conceptualization, S.X.; data curation, S.X.; formal analysis, S.X.; funding acquisition, X.S.; methodology, S.X.; project administration, X.S.; software, S.X.; supervision, X.S.; validation, S.X. and Z.X.; visualization, S.X.; writing—original draft, S.X.; writing—review and editing, X.S., S.X., Z.X. and X.H.

**Funding:** This work was supported by the National Nature Science Foundation of China under Grant No. 61473066 and No. 61601109, and the Fundamental Research Funds for the Central Universities under Grant No. N152305001.

**Acknowledgments:** The authors thank the anonymous reviewers for their insightful comments that helped improve the quality of this study.

**Conflicts of Interest:** The authors declare no conflict of interest.

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
