# Peer review of "Joint Optimal Power Allocation and Relay Selection Scheme in Energy Harvesting Two-Way Relaying Network"

_futureinternet, doi:10.3390/fi11020047_

Round 1

Reviewer 1 Report

1.       The performance of the proposed scheme is proved by simulating examples, however, how the simulation is working and what is the structure of that network are missing in this paper.

2.     In real situation, how the switch is carried out maybe a challenge due to the requirement of stability and reliability. Please try to explain in detail. 

Author Response

Point 1: The performance of the proposed scheme is proved by simulating examples, however, how the simulation is working and what is the structure of that network are missing in this paper.

Response 1:

We first compare the proposed scheme with the joint optimal power allocation and relay selection PS scheme and joint equal power allocation and random relay selection scheme. For the joint optimal power allocation and relay selection PS scheme, the instantaneous SNRs at two source nodes can be formulated as

And whereis the PS factor and it is fixed as 0.5. The max-min bidirectional information rate has the maximum when. Therefore, the optimal power allocation scheme under the total power constrain can be expressed as

Then we select cooperative relay node from the relay set, which can maximize the information rate. For the equal power allocation scheme, two source nodes have equal transmission powerand the relay selection is random.

Then we compare the proposed scheme with the joint power allocation and relay selection scheme in reference [20] and the joint equal power allocation and random relay selection scheme. From reference [20], the optimal values of,andcan be obtained as

With the optimal power allocation solution found, selecting the relay node with maximum SNR as cooperative relay. The traditional EPA scheme is defined asand the relay selection is random.

We have added some specific simulation parameters as well as the specifics of the comparison scheme in the new manuscript.

Point 2: In real situation, how the switch is carried out maybe a challenge due to the requirement of stability and reliability. Please try to explain in detail. 

Response 2:

We acknowledge that switching between two status maybe a challenge and require a strict synchronization process in real situation because the switch-like structure need to switch between receiving energy or receiving information status within a specific time. It means that we design how long time for energy harvesting and how long time for information reception is optimal. However, how the hardware switch is beyond the scope of our research. From the reference [1], that paper investigate a low-complexity antenna switching between decoding/rectifying in order to achieve simultaneous information/energy transfer. The proposed technique does not require any additional hardware or any specific synchronization process. Therefore, related technologies can effectively provide a stability and reliability energy harvesting and information reception. In addition, this paper is aiming to propose a joint power allocation and relay selection algorithm to improve the performance in two-way TS relay network.

[1]Krikidis, I.; Sasaki, S.; Timotheou, S.; Ding, Z. A Low Complexity Antenna Switching for Joint Wireless Information and Energy Transfer in MIMO Relay Channels. IEEE Transactions on Communications 2014, 62, 1577-1587.

All co-authors thank the anonymous reviewers for their insightful comments that helped improve the quality of this study.

Reviewer 2 Report

This paper propose a joint power allocation and relay selection algorithm based on TS relay architecture to maximize max-min bidirectional instantaneous information rate with simulation studies. The paper is well written. However, the system model seems very simple, and the conditions of simulation studies are not clearly descried. My concerns are:

1.      In section 1 “Introduction”, some related investigations are introduced. However, the research question, namely the problems identified are not properly summarized.

2.      Page 2, line 65-67, “At present, relay nodes adopt PSR protocol to achieve simultaneous wireless transmission of information and energy. However, the hardware of PSR protocol is difficult to realize. Thus we use TSR scheme ”. The authors’ reason to choose PSR has confused me. In my opinion, since PSR is the mainstream, it should not be something too difficult to realize.

3.      The conditions for simulations studies are not described in details, which are critical for the evaluation of the proposed solutions. Especially, when the simulation results are compared with traditional methods.

4.      Language is not too bad. But there are a lot of problems. Please improve the quality of English expressions. Some problems for instance:

Line 89, 93, 104, and 141 donate? should be denote?

Line 42-43, In [8], the one-way and two-way approach is based on …

Line 51, 79, etc. what is EH phase? Energy harvest?

Line 263, “superior to that the other two schemes”

Author Response

Point 1: In section 1 “Introduction”, some related investigations are introduced. However, the research question, namely the problems identified are not properly summarized.

Response 1: This paper has three contributions: first, we combine the time switching scheme in two-way relay network, which can solve the energy constraint problem of relay nodes; second, we proposed a joint relay selection and power allocation algorithm to maximize max-min instantaneous information rate and make up the loss of information transmission in extra time slots for energy transmission. The original non-convex problem is effectively solved; third, we compare the proposed scheme with several basic power allocation and relay selection schemes and PS scheme. The performance of two-way energy harvesting relay system has a significant improvement. The revised manuscript has concluded above contribution.

Point 2: Page 2, line 65-67, “At present, relay nodes adopt PSR protocol to achieve simultaneous wireless transmission of information and energy. However, the hardware of PSR protocol is difficult to realize. Thus we use TSR scheme”. The authors’ reason to choose PSR has confused me. In my opinion, since PSR is the mainstream, it should not be something too difficult to realize.

Response 2:

Currently, most references choose PSR scheme because it can achieve better performance theoretically. PSR means that the receiver splits the signal into two streams of different power for forwarding information and harvesting energy. The receiver architecture of PSR and TSR are shown in the following figure. As shown in the figure, the PSR receiver can achieve energy and information reception simultaneously. However, the PSR receiver requires appropriate power split circuits to decide how much of the signal is harvested as energy and how much of it is used for extracting information, which increase the complexity/cost of the hardware. It is difficult to realize and control. On the other hand, hardware non-idealities can significantly reduce the efficiency of the PS technology. While the TSR receiver is widely investigated due to its low implementation complexity. This proposal is mentioned in the following literature. Besides, TSR receiver needs to use addition phase for energy harvesting, which will reduce the instantaneous information rate. Thus, the power allocation, relay selection and related resource allocation schemes are investigated to make up the loss.

Ni, Z.; Motani, M. Transmission schemes and performance analysis for time-switching energy harvesting receivers. In Proceedings of IEEE International Conference on Communications.

Krikidis, I.; Sasaki, S.; Timotheou, S.; Ding, Z. A Low Complexity Antenna Switching for Joint Wireless Information and Energy Transfer in MIMO Relay Channels. IEEE Transactions on Communications 2014, 62, 1577-1587.

Point 3: The conditions for simulations studies are not described in details, which are critical for the evaluation of the proposed solutions. Especially, when the simulation results are compared with traditional methods.

Response 3:

We first compare the proposed scheme with the optimal power allocation and relay selection PS scheme and equal power allocation and random relay selection scheme. For the optimal power allocation and relay selection PS scheme, the instantaneous SNRs at two source nodes can be formulated as

And whereis the PS factor and it is fixed as 0.5. The max-min bidirectional information rate has the maximum when. Therefore, the optimal power allocation scheme under the total power constrain can be expressed as

Then we select cooperative relay node from the relay set, which can maximize the information rate. For the equal power allocation scheme, two source nodes have equal transmission powerand the relay selection is random.

Then we compare the proposed scheme with the joint power allocation and relay selection scheme in reference [20] and the equal power allocation and random relay selection scheme. From reference [20], the optimal values of,andcan be obtained as

With the optimal power allocation solution found, selecting the relay node with maximum SNR as cooperative relay. The traditional EPA scheme is defined asand the relay selection is random.

We have added some specific simulation parameters as well as the specifics of the comparison scheme in the new manuscript.

Point 4: Language is not too bad. But there are a lot of problems. Please improve the quality of English expressions. Some problems for instance:

Line 89, 93, 104, and 141 donate? should be denote?

Line 42-43, In [8], the one-way and two-way approach is based on …

Line 51, 79, etc. what is EH phase? Energy harvest?

Line 263, “superior to that the other two schemes

Response 4:

Line 89, 93, 104, and 141 should be modified as be denoted.

Line 42-43, the statement has changed: For maximizing the energy efficiency, the optimal power allocation and relay selection scheme can be obtained [22].

Line 51, 79, etc. EH is the abbreviation of the energy harvesting.

Line 263. From figure 6, it can be seen that the proposed scheme has better performance than the other two schemes.

The new manuscript will revise these issues and check the language grammar.

All co-authors thank the anonymous reviewers for their insightful comments that helped improve the quality of this study.

Reviewer 3 Report

The article describes a problem in power allocation for wireless communications and, as such, operates at the physical layer (L1). The journal that the article is submitted to is on "Future Internet" and as such, exists at L2 and above. It is unclear what the impact of the proposed physical layer power allocation algorithm is upon the design of networks and, more specifically, "Inter" networks. Certainly, the authors have not elaborated on this connection and it appears that they have submitted the article to a wrong journal.

They should should be encouraged to submit to a journal that specializes on physical layer research.

Author Response

Point 1: The article describes a problem in power allocation for wireless communications and, as such, operates at the physical layer (L1). The journal that the article is submitted to is on "Future Internet" and as such, exists at L2 and above. It is unclear what the impact of the proposed physical layer power allocation algorithm is upon the design of networks and, more specifically, "Inter" networks. Certainly, the authors have not elaborated on this connection and it appears that they have submitted the article to a wrong journal.

Response 1:

Energy harvesting is a promising technology, which can solve the energy constraint problem of the transmission node. At present, power splitting (PS) and time switching (TS) are two kinds of energy harvesting receiver architecture. Due to theoretically better performance, PS is widely used. However, the PS receiver needs to split the power in two different streams. And the receiver must decide how much of the signal is harvested as energy and how much of it is used for extracting information. It is difficult to realize and control. Therefore, we redesign a joint power allocation and relay selection scheme to maximize max-min bidirectional information rate when we exploit the TS scheme as a background. General speaking, TS scheme is a background. Our goal is to design an efficient resource allocation algorithm to improve system performance. The system model is simplified. In some actual scene, two users or sources in different cells want to exchange their own information through a power constrain relay node. The system model can be applied in this scene. In addition, the proposed algorithm and the system model can be widely used in 5G, wireless sensor, internet of things, cognitive radio and wireless body area network.

In addition, I found that a related relay selection article was published in "Future Internet". Please consider my manuscript.

Sahajwani. Manish, Jain. Alok and Gamad. Radheyshyam, “Log likelihood ratio based relay selection scheme for amplify and forward relaying with three state Markov channel, ” 2018, 10(9), 87.

All co-authors thank the anonymous reviewers for their insightful comments that helped improve the quality of this study.

Round 2

Reviewer 1 Report

The revised version is improved a lot and fine enough for publication. 

Reviewer 3 Report

The authors have not adequately addressed the concern that this paper is not "a networking paper". Rather, the authors have responded by saying that "energy harvesting" and "physical layer" methods will be useful in future wireless systems, such as 5G.

While that may be true, it is essentially the same as saying "improved modulation formats will be useful in future communication systems." A true statement, but does not at all address the networking versus physical layer aspects in communication system design. The whole point behind layered system design is to separate out aspects of receiver design from aspects of inter-networking. 

The reviewer found the rebuttal to lack a cogent argument supporting the admission of this paper into a journal devoted to networking technologies. To state that the journal accepted another similar paper is a flawed argument and in fact suggests that the other paper should not have been accepted.

In order for this paper to be considered in this journal, the authors must do more than write a physical layer paper that could be useful in future communication systems, but MUST address network aspects. This means that the authors must specifically identify network architectures or protocols that would utilize their "lower layer" research. This is possible, and would touch upon cross-layer design. If the authors choose to go this direction, then the reviewer would be happy to admit such an article. 

That said, the reviewer also reminds the authors that it is their explicit responsibility to address the scope of the journal, provided here for your reference. Note, this implies making an active proof that energy harvesting and power allocation and relay selection can be tied to one of the topics through network architecture and protocols (rather than merely state "it will be useful for <fill in the blank>").

Scope

Macro-Area 1: Smart System Technologies and Architecture

Included topics are:

•    advanced communications network infrastructures
•    evolution of internet basic services
•    internet of things
•    netted peripheral sensors
•    industrial internet
•    centralized and distributed data centers
•    embedded computing
•    cloud computing
•    software defined network functions and network virtualization
•    cloud-let and fog-computing
•    big data, open data and analytical tools
•    cyber-physical systems
•    network and distributed operating systems
•    web services
•    semantic structures and related software tools
•    artificial and augmented intelligence
•    augmented reality
•    system interoperability and flexible service composition
•    smart mission-critical system architectures
•    smart terminals and applications
•    pro-sumer tools for application design and development
•    cyber security compliance
•    privacy compliance
•    reliability compliance
•    dependability compliance
•    accountability compliance
•    trust compliance
•    technical quality of basic services

Macro-Area 2: Smart Systems and Applications

Included topics are:

•    smart mobility and transportation systems
•    smart utility systems
•    smart energy systems
•    smart living places
•    smart public government systems
•    smart health-care systems
•    smart systems for public security and safety
•    smart social assistance systems
•    smart geo-information and environmental monitoring systems
•    smart information-communications-knowledge delivery social systems
•    smart learning systems
•    smart manufacturing lines
•    smart liquid-enterprises
•    smart financial, payments and insurance systems
•    smart leisure systems
•    smart systems for cultural heritage conservation and fruition
•    smart city
•    application of new socio-economic systemic models for net-Living

Macro-Area 3: Net-Living Human Factors and Quality of Life enhancement

Included topics are:

•    human-computer interaction and usability
•    subjective human and social factors for well-being through Net-Living
•    end-user-centred design 
•    end-user constructive pro-activity enabling
•    social inclusion and cohesion enabling approaches
•    net-living lyfestyling personalization and optimization
•    quality of experience
•    living-labs
•    basic and vocational net-living education